Baleen whale inhalation variability revealed using animal-borne video tags

Nazario Emily C. enazario@ucsc.edu 1
Cade David E. 2
Bierlich K.C. 3
Czapanskiy Max F. 2
Goldbogen Jeremy A. 2
Kahane-Rapport Shirel R. 4
van der Hoop Julie M. 5
San Luis Merceline T. 1
Friedlaender Ari S. 6
1 Ecology and Evolutionary Biology, University of California, Santa Cruz , Santa Cruz , CA , United States of America
2 Department of Biology, Hopkins Marine Station, Stanford University , Pacific Grove , CA , United States of America
3 Marine Mammal Institute, Hatfield Marine Science Center, Oregon State University , Newport , OR , United States of America
4 Department of Biological Science, California State University, Fullerton , Fullerton , CA , United States of America
5 Zoophysiology at the Department of Bioscience, Aarhus University , Aarhus , Denmark
6 Institute of Marine Sciences, University of California, Santa Cruz , Santa Cruz , CA , United States of America
Ward Eric
Electronic publication date: 2022 Jul 20
Publication date: 2022
Volume: 10
Electronic Location ID: e13724
Received 2022 May 5; Accepted 2022 Jun 22
Copyright: ©2022 Nazario et al.
Copyright year: 2022
Copyright holder: Nazario et al.
License: This is an open access article distributed under the terms of the Creative Commons Attribution License, which permits unrestricted use, distribution, reproduction and adaptation in any medium and for any purpose provided that it is properly attributed. For attribution, the original author(s), title, publication source (PeerJ) and either DOI or URL of the article must be cited.
License URL: https://creativecommons.org/licenses/by/4.0/

Keywords: Ventilation, Nares, Rorqual whales, Biotelemetry

Funding: National Science Foundation OPP 1643877 National Science Foundation IOS-1656676 IOS-1656656 OPP-1644209 07-39483 Stanford University, the Myers Foundation American Cetacean Society San Francisco Bay Ethel M. Myers Oceanographic and Marine Biology Trust Funding for the Antarctic minke whale research was provided by the National Science Foundation OPP Award 1643877 to Ari S. Friedlaender. This research was also funded by grants from the National Science Foundation (IOS-1656676 and IOS-1656656; OPP-1644209 and 07-39483); the Office of Naval Research (N000141612477); a Terman Fellowship from Stanford University, the Myers Foundation, the American Cetacean Society and San Francisco Bay chapters, and the Dr. Earl H. Myers and Ethel M. Myers Oceanographic and Marine Biology Trust. The funders had no role in study design, data collection and analysis, decision to publish, or preparation of the manuscript.

==============================
Empirical metabolic rate and oxygen consumption estimates for free-ranging whales have been limited to counting respiratory events at the surface. Because these observations were limited and generally viewed from afar, variability in respiratory properties was unknown and oxygen consumption estimates assumed constant breath-to-breath tidal volume and oxygen uptake. However, evidence suggests that cetaceans in human care vary tidal volume and breathing frequency to meet aerobic demand, which would significantly impact energetic estimates if the findings held in free-ranging species. In this study, we used suction cup-attached video tags positioned posterior to the nares of two humpback whales (Megaptera novaeangliae) and four Antarctic minke whales (Balaenoptera bonaerensis) to measure inhalation duration, relative nares expansion, and maximum nares expansion. Inhalation duration and nares expansion varied between and within initial, middle, and terminal breaths of surface sequences between dives. The initial and middle breaths exhibited the least variability and had the shortest durations and smallest nares expansions. In contrast, terminal breaths were highly variable, with the longest inhalation durations and the largest nares expansions. Our results demonstrate breath-to-breath variability in duration and nares expansion, suggesting differential oxygen exchange in each breath during the surface interval. With future validation, inhalation duration or nares area could be used alongside respiratory frequency to improve oxygen consumption estimates by accounting for breath-to-breath variation in wild whales.

Introduction

Understanding the physiology and bioenergetics of an organism is fundamental for gaining insight into the caloric requirements for thriving in its environment (Speakman, 1999). Primary methods for quantifying energetics include directly measuring or estimating an organism’s daily energetic budget or the energetic costs of specific behaviors (Nagy, 1987; Speakman, 1999; Williams et al., 2004). Energetic costs can be measured using techniques such as doubly labeled water, respirometry, and (with proper calibration) tri-axial accelerometers (Davis, Williams & Kooyman, 1985; Speakman & Król, 2005; Wilson et al., 2006; Shaffer, 2011). While these methods have provided valuable information regarding energy expenditure and metabolic costs for a wide variety of species, these approaches are typically unavailable or unrealistic for large, free-ranging, and/or cryptic species.

Performing the above techniques on marine vertebrates, such as pinnipeds and cetaceans, is particularly challenging due to their large size and remote habitats. While direct measurements and estimations have been conducted on pinnipeds and small cetaceans in human care (Liwanag et al., 2009; Williams & Noren, 2009; Williams et al., 2017), other methods have been developed for free-ranging species. As cetaceans must return to the surface to replenish their oxygen stores and offload carbon dioxide, recording breathing frequency has been a common method of studying baleen whale oxygen consumption (and therefore energetics) due to the simplicity of counting the distinct exhalations (Sumich, 1983; Dolphin, 1987a; Blix & Folkow, 1995; Goldbogen et al., 2008; Christiansen, Rasmussen & Lusseau, 2014;). However, using breathing frequency as a proxy for oxygen consumption requires the assumption that other components of respiratory physiology (tidal volume and oxygen extraction) do not substantially vary breath-to-breath (Sumich, 1983; Blix & Folkow, 1995; Goldbogen et al., 2008; Christiansen, Rasmussen & Lusseau, 2014; Fahlman et al., 2016; Cauture et al., 2019). While tidal volume was assumed to be constant during energetically costly behaviors, e.g., during exercise (Yazdi, Kilian & Culik, 1999), evidence suggests that tidal volume varies with activity in both cetaceans and pinnipeds (Kooyman et al., 1971; Fahlman et al., 2016). Studies with bottlenose dolphins in human care, free-ranging killer whales (Orcinus orca), and free-ranging minke whales have demonstrated that fixed oxygen uptake values led to weak relationships between oxygen consumption and activity levels (Williams and Noren, 2009; Christiansen, Rasmussen & Lusseau, 2014; Fahlman et al., 2016; Roos, Wu & Miller, 2016). This deviation could be explained by tidal volumes being consistently lower than the total lung capacity, as well as variation in tidal volume in response to activity level (Kooyman et al., 1971; Fahlman, Moore & Garcia-Parraga, 2017). Additional work with bottlenose dolphins in human care showed that considerable breath-to-breath variation has direct effects on gas-exchange and estimates of energetic requirements (Fahlman et al., 2016). Due to the difficulty of taking these measurements in the field, especially for large whales, additional methods are needed to estimate if and how these variations in breath volume occur in free-ranging whales.

In this study, we developed a novel method to estimate breath-to-breath variability throughout a surface interval in free-ranging rorqual whales. In cetaceans, tidal volume (m3) is the product of three physical properties: inhalation duration (s), area of the nares opening (m2), and flow velocity (m s−1). Using animal-borne cameras, we measured the inhalation duration and relative area of nares expansion. Previous studies have observed that cetacean respiratory flow reaches and maintains near-maximum values for almost the entire inhalation, regardless of lung volume (Kooyman & Sinnett, 1979; Kooyman & Cornell, 1981; Fahlman et al., 2019a). Our method therefore presumes constant respiratory flow velocity, which approximates observed respiration patterns characterized by near-maximal flow for almost the entire inhalation (Sumich, 2001; Fahlman et al., 2015; Fahlman et al., 2019a). We measured the duration of inhalation events and the magnitude of nares expansion during inhalations to evaluate the breath-to-breath variability in free-ranging rorqual whales. We hypothesized that inhalation duration, integrated nares area (i.e., nares area integrated over time during the inhalation), and maximum nares area would vary between and within breaths completed throughout a surface interval. Additionally, we hypothesized that the magnitude of the total inhalation duration (the total amount of time spent inhaling for a given surface interval) of a surface sequence would reflect the dive effort (duration, feeding rate) of the previous dive, indicating that the whale varies the duration of the inhalation according to its recovery requirements. Similarly, we expected that the inhalation durations of a surface sequence would also reflect the dive effort of the upcoming dive, which could indicate that the whale is varying its inhalation patterns in anticipation of dive activity.

Materials & Methods

Data collection

This study used data acquired through past projects investigating the underwater behavior of free-ranging rorqual whales (Goldbogen et al., 2019; Kahane-Rapport et al., 2020; Savoca et al., 2021). This data was an aggregation of deployments using video-recording motion sensor tags from Customized Animal Tracking Solutions (CATS; Table 1). Deployments were selected if the tags were placed directly behind and facing the nares, the nares were clearly visible in the video data, and there was at least 2 h of video where the tag did not shift position. A single reviewer recorded the data from each of the videos and confirmed that the camera position remained the same throughout the observed portion of the deployment. We used data from two humpback whale (Megaptera novaeangliae) deployments in Monterey Bay, CA, and four Antarctic minke whale (Balaenoptera bonaerensis) deployments from the Western Antarctic Peninsula (Table 2). The humpback tag deployments were in the Monterey Bay National Marine Sanctuary under the NMFS permit #20430. The Antarctic minke whale tags were deployed in the West Antarctic peninsula under NMFS permit #23095. This study was also conducted under the UCSC IACUC permit Friea1706 and the ACA permit 2015-011. In both locations, whales were approached obliquely in a small boat and a suction-cup attached CATS tag was placed on the animal using a carbon-fiber pole (Friedlaender et al., 2009), posterior to and facing the nares. The archival CATS tags are equipped with tri-axial accelerometers, magnetometers, pressure sensors, gyroscopes, and a time-depth recorder as well as a video camera with 1, 920 × 1, 080 or 1, 280 × 720 resolution, and a frame rate of 30 Hz (Cade et al., 2021). The frame rate of 30 Hz set the temporal resolution of our video analysis. Tag accelerometers were sampled at 400 Hz, magnetometers and gyroscopes at 50 Hz, and pressure and light at 10 Hz, then all data were decimated to 10 Hz. Tag orientation on the animal was corrected to calculate animal orientation (i.e., pitch, roll & heading), and data were time-synchronized with video using embedded time stamps on the videos (Cade et al., 2021). The video cameras were programmed to record based on light level and were generally active when the whales were near or at the surface, allowing for continuous video to be collected from surface series. To recover the tags, we used an Argos satellite transmitter and a VHF beacon.

Table 1 List of symbols and abbreviations.

This list of symbols and abbreviations represent the terms used through this research to describe and support our method and conclusions.

CATS	Customized Animal Tracking Solutions	
BORIS	Behavioral Observation Research Interactive Software	
IBI	Inter-Breath-Interval	
IA	Integrated Area	
GLMM	Generalized Linear Mixed Effects Model	

Table 2 Whale morphometrics.

Summarizes the morphometrics for each humpback and Antarctic minke whale used in this study. Morphometric information includes physical maturity, sexual maturity, and mean length. We have also included animal location.

Individual ID	Sexual maturity	Physical maturity	Mean total length (m)	Location	
mn180607-44	N/A	N/A	N/A	Monterey, CA	
mn161117-10	Sexually Mature	N/A	9.71	Monterey, CA	
bb190228-55b	Sexually Immature	Physically Immature	7.06	Western Antarctic Peninsula	
bb190309-52	Sexually Immature	Physically Immature	7.16	Western Antarctic Peninsula	
bb180304-45	Sexually Mature	Physically Immature	8.58	Western Antarctic Peninsula	
bb190224-52	Sexually Immature	Physically Immature	4.54	Western Antarctic Peninsula	

Video analysis and integrated area calculations

The videos were examined in Behavioral Observation Research Interactive Software (BORIS; Friard & Gamba, 2016) and inhalations were identified when the nares broke the surface and began expanding (Fig. 1). Exhalations regularly started below the surface and long bubble streams often obstructed direct observation of the nares. Inhalations were consistently visible at the surface, allowing for more reliable measurements of area and duration. To distinguish between an exhalation and an inhalation at the surface, we saw that the nares were consistently narrower for an exhalation and the nares rapidly expanded for an inhalation (Movie S1). Given the distinct change in nares shape as the breath transitioned from an exhalation to an inhalation and the use of frame-by-frame mode in BORIS, we were confident in our ability to determine the exact start of each inhalation. We only used video footage with a clear view of the nares unobstructed by splashing water or glare. A surface sequence was defined as a series of breaths between dives. We classified submersions longer than 90 s a dive (Table 3). Submersions that were shorter than or equal to 90 s were classified as an inter-breath-interval (IBI). To identify the appropriate value that should be used to distinguish an IBI and a dive, we found the intermediate mean of the log transformed bimodal distribution of submergence durations. The first surface sequence analyzed was at least 30 min after the tag was deployed to reduce any variability introduced by tag deployment related stress and to ensure the tag remained in a stable position (Jahoda et al., 2003). We recorded the elapsed time, depth, speed, and jerk (the norm of the difference of the 10 Hz accelerometer signal) of the dive immediately before and immediately after each inhalation (Simon, Johnson & Madsen, 2012). Speed and jerk were used to identify feeding lunges, which were manually identified as increases in speed (up to 3 to 4 m s−1) followed by rapid deceleration coincident with mouth opening, with simultaneous peaks in jerk (20 to 30 m s−3) (Cade et al., 2016).

Figure 1 Frames measuring integrated area over time.

(A) This image sequence depicts our method for calculating the integrated area of the nares for humpback whale mn180607-44. The area of the pixels inside the yellow circle were measured for each frame (represented in text). (B) The plot depicts how the frame area changes over the duration of the inhalation. The y-axis was normalized to the largest area for each sequence of frames and the x-axis normalized to the longest duration for each sequence of frames.

Table 3 The summary statistics for dives and inter-breath-intervals (IBI).

The minimum, mean ±SD, maximum, and count for dives and IBIs are presented here for each whale. The individual ID is the field ID given to each whale.

Dive and IBI Submergence Duration Summary Statistics	
Individual ID	Dive or IBI	Minimum (s)	Mean ± SD (s)	Maximum (s)	n	
bb180304-45	Dive	130	253.75 ± 99.267	380	8	
bb190224-52	96	124.125 ± 18.980	162	16	
bb190228-55b	195	232.714 ± 32.153	325	21	
bb190309-52	98	337.923 ± 119.161	510	13	
mn161117-10	92	254.057 ± 92.101	462	35	
mn180607-44	176	300.790 ± 69.292	430	19	
bb180304-45	IBI	9	27.136 ± 18.964	90	59	
bb190224-52	3	21.344 ± 20.861	84	131	
bb190228-55b	6	17.310 ± 8.648	88	116	
bb190309-52	6	16.600 ± 8.940	48	95	
mn161117-10	5	19.435 ± 8.055	68	108	
mn180607-44	6	14.649 ± 9.262	71	111	

Breaths were categorized as either an initial breath (the first breath after a dive), middle breath (all breaths between the first breath and the last breath of a surface sequence were labelled as middle breaths), or terminal breath (the last breath before a dive). Single breath events were omitted due to their infrequency (n < 5).

Inhalations were grouped by surface sequence. Inhalation duration, tag position on the animal, maximum dive depth before and after the surface sequence, duration of the submergence before and after the surface sequence, and the number of lunges on the dive immediately before or immediately after the surface sequence were also recorded for each inhalation. The beginning and end of an inhalation were determined by the first and last video frames displaying nares expansion or opening, respectively, and every frame between the start and end were exported as images with their associated timestamp. We assumed nares expansion was synchronous across the left and right nares and used either the left or right nares only, based on camera angle, for the entire deployment’s measurements. We confirmed this assumption by randomly selecting two whales from our total sample size and ran a paired two sample t-test which compared the left nares area to the right nares area (area calculations described below). Prior to running the paired t-test we confirmed normality and homogeneity of variance. Our results suggest that there is no significant difference in the integrated area measurements between the left and right nares (paired two sample t-test, df = 29 observations per whale, whale one p = 0.18, whale two p = 0.74).

To quantify the integrated area (IA; Table 4) of nares opening, the frame sequences were examined in ImageJ 1.52a (Schneider, Rasband & Eliceiri, 2012) and the area of the region inside the nares was measured in pixels (Fig. 1). Given the unique camera placement on each whale and the corresponding distortion that then influences the area measurements used to calculate the IA, it is important to note that the raw IA and maximum area values are not absolute. Instead, the IA and maximum area for each breath are relative values for each whale deployment that cannot be compared across individuals. To compare both metrics across whales, we independently normalized the IA and maximum nares expansion to the maximal value measured for each whale, respectively. IA was determined as the sum of the mean area of each two consecutive frames multiplied by the time between the frames as follows (for n frames in a sequence where A is area and t is time; Eq. (1)): (1) IA= ∑i=1n−1Ai+Ai+12×ti+1−ti.

Table 4 Inhalation duration, normalized IA, and normalized maximum nares area means and standard deviations.

The mean and standard deviation results for the inhalation duration, normalized IA, and maximum nares area across breath types and individual whales. Rows are grouped by whale field ID, and columns specify the breath type and metrics. The sample size, n, in this table represents the number of observations included for that whale and breath type for the inhalation duration, IA, and maximum area.

The Mean and Standard Deviation (SD) for the Inhalation Duration, Normalized IA, and Normalized Maximum Nares Expansion Across Breath Types and Individuals	
Individual ID	Breath type	Mean inhalation duration (s)	SD inhalation duration (s)	n	Mean normalized IA (px)	SD normalized IA (px)	n	Mean maximum area (px)	SD maximum area (px)	n	
bb180304-45	Initial	0.46275	0.195793441	8	0.197062714	0.177507143	6	0.476726727	0.231119567	6	
bb190224-52	Initial	0.345818182	0.144029037	11	0.439275208	0.276310777	6	0.544701987	0.189664791	8	
bb190228-55b	Initial	0.708952381	0.126232514	21	0.35470339	0.068382001	20	0.535303329	0.11512015	19	
bb190309-52	Initial	0.432615385	0.184307957	13	0.193241027	0.165589675	11	0.290991176	0.107674965	11	
mn161117-10	Initial	0.95328125	0.222537744	32	0.174283575	0.104739032	21	0.376554174	0.166162437	21	
mn180607-44	Initial	0.879105263	0.240920202	19	0.157982088	0.090745624	17	0.384435157	0.147198695	14	
bb180304-45	Middle	0.55178	0.193578681	50	0.320608611	0.205368257	43	0.562853551	0.163254678	43	
bb190224-52	Middle	0.312409091	0.138805631	88	0.365364046	0.261786608	64	0.542455061	0.21167685	56	
bb190228-55b	Middle	0.700705263	0.09656217	95	0.328226525	0.073211764	78	0.505286762	0.093443243	75	
bb190309-52	Middle	0.486682927	0.159863899	82	0.269220556	0.177544644	79	0.360145155	0.179856425	79	
mn161117-10	Middle	1.082943662	0.278280428	71	0.276174755	0.132990822	54	0.507191864	0.161930637	51	
mn180607-44	Middle	0.92021875	0.173415728	96	0.166016634	0.086524258	86	0.401898206	0.156483772	85	
bb180304-45	Terminal	0.722666667	0.240740005	9	0.517032003	0.288649005	8	0.832207207	0.159998254	8	
bb190224-52	Terminal	0.390944444	0.122697367	18	0.474228382	0.216233523	8	0.65607064	0.116855497	15	
bb190228-55b	Terminal	0.897809524	0.096075813	21	0.707775602	0.169561388	20	0.786545925	0.169976868	19	
bb190309-52	Terminal	0.647833333	0.107691338	12	0.586829963	0.216212763	12	0.66755592	0.221893042	11	
mn161117-10	Terminal	1.81571875	0.350399057	32	0.445958148	0.241859946	24	0.618185544	0.160771403	26	
mn180607-44	Terminal	1.551933333	0.297869211	15	0.541828933	0.224848797	15	0.781983088	0.140306131	13	

The maximum nares expansion was recorded by taking the maximum area measured for each inhalation. IA and maximum area were measured in pixels, and the IA was multiplied by the duration in seconds and represents the relative magnitude of total nares expansion throughout the duration of the inhalation. Using the Antarctic minke whale data, we determined whether there were any differences between IA measurements that used every frame of an inhalation sequence as opposed to every other frame of an inhalation sequence. We compared a random subsample of breaths for both minke whales and ran a paired two-tailed t-test for both minke whales after confirming normality and homogeneity of variance. Although the IA measurements were significantly different between the two methods for both minke whales (paired two-tailed t-test, df = 12 observations per whale, whale one p = 0.004, whale two p = 0.007), the IA values that were calculated using every frame were only 5% greater on average than the IA values calculated using every other frame (Fig. S1). We concluded this difference to have negligible effects on the interpretation of the results given that the IA is a relative value. Due to the long inhalation durations of humpback whales, every other frame of the inhalation frame sequence was used. Despite measuring the area of a nares on every other frame, this still accounted for the total duration of the inhalation and the overall degree of nares expansion. For Antarctic minke whales all frames were used due to their shorter inhalation durations. For both whale species, regardless of the method used, we calculated the IA using between 10-20 frames for each inhalation.

Statistical analysis

IA measurements were calculated in R (R Core Team, 2019) using packages from the tidyverse suite (Wickham et al., 2019; Aphalo, 2019). All analysis was performed in R.

Inhalation duration, integrated area, and maximum area

The distance of the camera from the nares, the proportion of the nares that was visible, and the angled view of the nares varied for each whale. To account for this the IA was normalized to the maximum IA of each whale to make graphical comparisons across individuals possible, and each whale was separately grouped for all statistical tests. All IA reported are the percentage of the maximum for each whale. Maximum nares expansion was also normalized to the maximal value for each whale for similar reasons described for IA. For the inhalation duration, IA, and maximum nares area of each breath type normality was confirmed but homogeneity of variance was violated. Therefore, we ran a maximum likelihood generalized linear mixed effects model (GLMM) which is more robust to heteroscedasticity. For the IA and maximum nares area we fit a GLMM using the beta distribution and a logit link function. For the inhalation duration we fit a GLMM using the gamma distribution and a log link function (Brooks et al., 2017). Breath type was the predictor variable and whale ID was a random effect for each of the models. We then ran a post-hoc contrast test to compare each breath type pair for the inhalation duration, IA, and maximum nares expansion. We used a Bonferroni-adjusted critical p-value for multiple hypothesis comparisons. To evaluate the relationship between maximum nares area and inhalation duration we fit a GLMM with beta distribution and a logit link function as previously described, but instead inhalation duration was the predictor variable and maximum nares area was the response. Additionally, species was added as a fixed effect and whale ID was kept as a random effect. The critical p-value remained as 0.05. For all confidence intervals calculated for each GLMM described previously and hereafter we used the Wald method.

Inhalation duration and breath count vs. dive parameters

To determine the relationship between ventilation and the previous and following dive’s duration and lunge count, we fit a second degree polynomial GLMM using the gamma distribution and a log link function. Here, the predictor variables were dive duration and lunge count, and the response variable was the total inhale duration for each surface interval. The total inhalation duration was calculated by summing each inhalation duration from each breath across all breaths in a surface sequence. Thus, giving the total amount of time the whale inhaled for that surface seqeunce. Whale ID and species type were added as random effects. IA and maximum nares area were not used as response variables as these are relative values that largely vary in response to camera placement. To compare total inhale duration and breath count, we fit another second degree polynomial GLMM using the Poisson distribution and a log link function. Here, the predictor variables were dive duration and lunge count from the previous or upcoming dive, and the response variable was the breath count from each surface sequence. We used a Bonferroni-adjusted critical p-value for multiple hypothesis comparisons. Additionally, after fitting each model, model validation was conducted by visually checking the residuals.

Figure 2 Normalized nares expansion over time.

Relationship between normalized inhalation duration and frame area during initial, middle, and terminal breaths. The transparent lines show all the breaths for a given animal and the brighter lines represent the moving average for that breath type. The grey line is the flow rate over time for an inhalation as displayed in Sumich (2001). The line pictured here, however, is plotted along a positive y-axis instead of a negative one as originally done in the article. The x-axis was normalized to the longest inhalation duration, the left y-axis was normalized to the largest frame area, and the right y-axis was normalized to the largest ûow rate. The labels for each column represent the species (bb = Antarctic minke whale, mn = humpback) and the field identification number.

Results

Inhalation duration, IA, and maximum area variability

The relationship between the frame area and inhale duration did not vary across breath types, this relationship did vary, however, between species (Fig. 2). The frame area of the Antarctic minke whales initially rapidly increased, briefly plateaued, and then rapidly decreased at the end of the inhalation. The frame area of the humpback whales rapidly increased, plateaued for a majority of the inhalation, then rapidly decreased. The rapid increase –plateau –rapid decrease pattern observed in the timing of nares expansion for both species was similar in shape to previously measured inspiratory flow rate curves from captive cetaceans (Kooyman & Cornell, 1981; Sumich, 2001; Sumich & May, 2009; Fahlman et al., 2015; Fahlman et al., 2019a).

The inhalation duration was longest for the terminal breath, moderate for the middle breath, and shortest for the initial breath. The terminal breath inhalation duration for both species was significantly longer than the initial (GLMM, post hoc contrast test, n = 6 whales, p < 0.0001) and middle breaths (GLMM, post hoc contrast test, n = 6 whales, p < 0.0001; Fig. 3; Table 4; Table S1). The middle breath inhalation durations not significantly longer than the initial breath inhalation durations (GLMM, post hoc contrast test, n = 6 whales, p = 0.054).

Figure 3 Inhalation duration, normalized integrated area, and normalized maximum nares area across breath types.

Boxplot showing (top) the inhale duration, (middle) the normalized integrated area, and (bottom) the normalized maximum nares expansion for initial, middle, and terminal breaths for all whales in the study. The box plots provide the mean, standard error, minimum, maximum, and outliers. The integrated and maximum nares area were normalized to the largest inhalation for each animal. The points on both plots represent outliers. Each whale is plotted separately according to their whale field ID and are represented by the different colors.

The initial breath normalized integrated area (IA) was small and the least variable relative to the other two breath types for both species (Table 4). Middle breaths were similar in size to initial breaths but were more variable for both species. Terminal breaths were the most variable and were significantly larger in IA than the initial (GLMM, post-hoc contrast test, n = 6 whales, p < 0.0001) and middle breaths for both species (GLMM, post-hoc contrast test, n = 6 whales, p < 0.0001; Fig. 3; Table S1). Initial and middle breath IAs were not significantly different (GLMM, post hoc contrast test, n = 6 whales, p = 0.128).

The maximum nares expansion showed similar trends across breath types as the IA. The maximum nares expansion for initial and middle breaths were not significantly different (GLMM, post hoc contrast test, n = 6 whales, p = 0.284; Fig. 3; Table 4; Table S1). Terminal breath maximum nares expansion was significantly larger than the initial breaths (GLMM, post hoc contrast test, n = 6 whales, p < 0.0001) and the middle breaths (GLMM, post hoc contrast test, n = 6 whales, p < 0.0001; Fig. 3). Inhalation duration had a significant positive correlation with maximum nares expansion (GLMM, n = 6 whales, p < 0.0001) and species type had a significant effect (GLMM, n = 6, p < 0.0001; Fig. 4; Fig. S2).

Figure 4 Normalized maximum nares area vs. inhalation duration.

Normalized integrated area plotted against inhalation duration separated by species. The black line represents the GLMM where inhale duration and species were the predictor variables and normalized maximum nares area was the response variable. Whale field ID was included as a random effect. Observations were grouped by breath type (initial, middle, or terminal). The labels for each panel represent the species (mn = humpback, bb = Antarctic minke whale). The y-axis was normalized to the largest maximum nares area for each individual.

Inhale duration, breath count, and associated dive parameters

The magnitude of the total inhalation duration was not significantly correlated with the previous dive’s duration (GLMM, n = 6 whales, p = 0.456; AIC = 456.4) or lunge count (GLMM, n = 6 whales, p = 0.604; AIC = 461.4; Table 5; Table S1). The best model, the one with the lowest AIC score, for the total inhalation duration versus the upcoming dive’s parameters used both the upcoming dive duration and lunge count as predictor variables and indicated a significant relationship (GLMM, n = 6 whales, dive duration p = 0.00630, lunge count p = 0.00103; AIC = 437.9; Fig. 5; Table 5; Table S1). The breath count was not significantly correlated with the previous (GLMM, n = 6, dive duration p = 0.610, lunge count p = 0.161; AIC = 504.5; Table 5; Fig. 5.; Table S1 ; Fig. S3) or upcoming dive’s duration and lunge count (GLMM, n = 6, dive duration p = 0.102, lunge count p = 0.318; AIC = 502).

Table 5 GLMM results for the total inhalation duration and breath count vs. the previous and upcoming dive duration and lunge count.

The p-values for the GLMMs run for each dive parameter predictor variable (dive duration and lunge count). The response variables were the total inhalation duration and breath count. The p-values were corrected for multiple hypotheses using a Bonferroni adjustment (total inhalation duration α = 0.0125; breath count α = 0.025) and significant values are in bold. The n for each model was 6 whales.

Metric	Dive parameter	Position relative to surface sequence	p-value	
Total Inhalation Duration	Dive Duration	Previous	0.456	
Total Inhalation Duration	Lunge Count	Previous	0.604	
Breath Count	Dive Duration & Lunge Count	Previous	Duration (0.610)
Lunge Count (0.161)	
Total Inhalation Duration	Dive Duration	Upcoming	0.00242	
Total Inhalation Duration	Lunge Count	Upcoming	0.00073	
Total Inhalation Duration	Dive Duration & Lunge Count	Upcoming	Duration (0.00630)
Lunge Count (0.00103)	
Breath Count	Dive Duration & Lunge Count	Upcoming	Duration (0.101)
Lunge Count (0.318)	

Figure 5 Total inhale duration and breath count vs. upcoming dive duration and lunge count.

The total duration spent inhaling per surface sequence plotted against the upcoming dive duration. The colored lines are separated by lunge count categories defined separately for humpback and minke whales. For minke whales: non-foraging = 0 lunges; moderate = 1–4 lunges; high = ≥ 5 lunges. For humpback whales: non-foraging = 0 lunges; moderate = 1–2 lunges; high = ≥ 3 lunges. No model was plotted for high lunges for humpback whales due to the small sample size. The lines represent the second degree polynomial GLMM where both lunge count and dive duration of the upcoming dive were predictor variables, total inhale duration or breath count was the response, and whale ID and species were random effects. Observations were separated by species. The labels for each panel represent the species (mn = humpback, bb = Antarctic minke whale).

Discussion

This study developed a new method using video data from suction cup-attached tags to evaluate the extent and duration of nares opening to evaluate the breath-to-breath variability of free-ranging rorqual whales in conjunction with dive behavior. Unlike previous studies, this is also the first detailed examination of how inhalations change throughout a surface sequence in free-ranging whales. We confirmed that inhalation duration, integrated area (IA), and maximum nares area, and thus the overall magnitude of each inhalation, vary with and between breath types in free-ranging rorqual whales. Our results did not find a significant correlation between the total inhalation duration and the previous dive’s level of activity (e.g., lunge filter feeding, dive duration), however inhalation duration was significantly correlated with the following dive’s duration and lunge count. Thus, with further validation, inhalation duration and nares expansion could be used to improve estimates of wild cetacean energetics relative to traditional methods alone (e.g., breath frequency).

By recording inhalation duration and quantifying nares expansion, we observed considerable breath-to-breath variation in the inhalation duration and surface of the nares area. This variation was greatest for the terminal breaths (Fig. 3). The terminal breath seems to be much larger than other breath categories, as the mean inhalation duration for terminal breaths was about two times larger than the initial breath, the mean terminal breath IA was over three times larger than initial breath IA, and the mean maximum nares area for terminal breaths was over two times larger than the mean maximum area for initial breaths (Table 4; Fig. 3). Large terminal breaths have also been identified in other diving animals, such as the Humboldt (Spheniscus humboldti) and Megellanic (Spehniscus magellanicus) penguins, and are associated with efficient blood and muscle oxygen loading in Emperor penguins (Aptenodytes forsteri; Wilson et al., 2003; Ponganis et al., 2009). It is likely that variations in tidal volume at this scale directly influence estimates of gas exchange and metabolic rate (Spencer, Gornall 3rd & Poulter, 1967; Fahlman et al., 2016; Roos, Wu & Miller, 2016).

We compared nares expansion over time to previously published flow rate measurements to substantiate the use of nares expansion as a reliable proxy for quantifying breath variation (Kooyman & Cornell, 1981; Sumich, 2001; Sumich & May, 2009; Fahlman et al., 2015). The timing of nares expansion we observed through animal-borne camera data was similar to that of inspiratory flow rates measured in a gray whale calf, bottlenose dolphins, and beluga whales in human care (Kooyman & Cornell, 1981; Sumich, 2001; Sumich & May, 2009; Fahlman et al., 2015; Fahlman et al., 2019a). Generally, both the flow rate and nares expansion curves increased, plateaued, and then rapidly decreased (Fig. 2). These previous measurements showed consistent inspiratory flow rates over a majority of the inhalation despite changing tidal volumes, which is comparable to the pattern of nares expansion over a majority of the inhalation (Kooyman & Cornell, 1981; Sumich, 2001; Sumich & May, 2009; Fahlman et al., 2015; Fahlman et al., 2019a;). This similarity in shape between the flow rate and nares expansion across breath types suggests that the timing of nares expansion is a reasonable proxy for relative flow rate, thus assuming that flow rate is constant over the inhalation and is related to nares expansion. The similarity across breath types also suggests that any variation between breaths is likely originating from other metrics such as inhalation duration or the magnitude of nares expansion. There is some variation, however, in nares expansion over time between species. Antarctic minke whales quickly expanded and contracted their nares minimizing the amount of time they remained open at the surface. While humpback whales sustained expanded nares for a longer proportion of the inhalation duration. Future validation experiments should thus be conducted to identify the relationship between flow rate, tidal volume, inhalation duration, and nares expansion and how these vary among species. Due to the relative nature of nares expansion, we used our method of nares expansion to assess the breath-to-breath variability within individuals, as has previously been measured in the bottlenose dolphin, Humboldt and Megellanic penguins, and grey seals (Reed et al., 1994; Wilson et al., 2003; Fahlman et al., 2016; Fahlman et al., 2019b).

Our data indicated that initial breaths had the shortest inhalation duration, smallest IA, smallest maximum nares expansion values, and most consistency for all metrics (Table 4). The middle breath inhalation durations were similar to initial breaths in area and duration, but were slightly more variable than the initial breaths (Fig. 3; Table 4). Inhalation duration, IA, and maximum nares expansion are all relatively variable throughout a surface interval which suggests some variation between middle breaths. Previous studies have found a close relationship between the number of middle breaths and the previous dive’s activity level, and given our results, middle breath variance may also reflect similar energetic costs of either the previous or upcoming dives (Dolphin, 1987b; Chu, 1988; Goldbogen et al., 2008).

Our results differ from previous studies measuring respiratory patterns in Humboldt and Megellanic penguins, where Wilson et al. (2003) found that the initial breath was one of the largest breaths taken in a surface interval. Large deep initial breaths would take advantage of the large oxygen and carbon dioxide partial pressure gradients between the air in the lungs and the venous blood following an energetically expensive dive (Craig Jr & Påsche, 1980). This large gradient increases the rate of gas diffusion which can be used to maximize gas exchange, as the gradient is strongest for the first few breaths of a surface series, and plateaus over time (Boutilier, Reed & Fedak, 2001). Despite our results suggesting that the initial breath was the smallest in IA and duration, the middle breath inhale duration and IA were observed to be more variable than the initial breaths. Thus, this may suggest that the animal may be taking advantage of the large partial pressure gradient by varying the number of middle breaths, as observed in previous studies, in addition to breath-to-breath changes in inhalation duration and IA (Sumich, 1983; Goldbogen et al., 2008; Christiansen, Rasmussen & Lusseau, 2014; Fig. 3). This difference in ventilation patterns may also be attributed to differences relating to avian vs. mammalian respiratory systems. Diving birds are much more dependent on respiratory oxygen reserves relative to marine mammals who primarily rely on oxygen stores in the blood and muscle (Ponganis, Meir & Williams, 2011).

Our data also indicated that the last breath before a dive had significantly longer inhalation durations, larger nares integrated area, and larger maximum nares areas, which suggests that the terminal breath is the largest breath within a surface interval (Fig. 3). The terminal breath was also more variable than the other breath types (Table 4). The variability across the ventilation metrics agrees with past measurements in bottlenose dolphins, and that each breath in a surface sequence is unique (Fahlman et al., 2016). Previous studies estimating body density, passive drag, and buoyancy in cetaceans at depth have also assumed diving gas volume variability (Miller et al., 2004; Miller et al., 2016; Narazaki et al., 2018). Thus, our results suggest that these assumptions are valid and that the diving gas volume likely varies dive to dive.

We found that the total inhalation duration had no significant correlation with the previous dive’s duration or lunge count. Previously published work, however, has found a positive relationship between the previous dive’s duration and animal’s behavior and the upcoming breath count (Goldbogen et al., 2008). Despite our results suggesting no relationship between the previous dive’s level of activity and the following duration spent inhaling, future studies should further examine the relationship between these additional ventilation metrics and diving behavior. Additionally, given our results suggesting no relationship between breath count and the previous or upcoming dive’s duration and lunge count, exploring the use of the inhalation duration technique could add value to breath counts alone (Fig. 5; Table 5). By measuring the nares expansion and durations of exhalations, in addition to inhalations, more information could be revealed regarding how ventilation may change following dives of varying durations and behaviors.

With respect to the upcoming dive duration and lunge count, we found that the whales took the longest breaths prior to dives of moderate durations and with moderate to high lunge counts (Fig. 5; Table 5). The total duration spent inhaling at the surface was similar in length before dives with low and high dive durations. These durations were also similar in length prior to dives with moderate and high lunge counts. Based off our results and because whales generally feed in prolonged bouts that include many consecutive foraging dives, we expect that they may plan for their foraging effort on an upcoming dive based on some information from the previous dive and could vary their respiratory oxygen stores via time spent inhaling accordingly (Hazen, Friedlaender & Goldbogen, 2015. When upcoming dives are low to moderate in duration or are either non-foraging or have moderate lunge counts, our results indicate that the whales adjust their total time spent inhaling (Fig. 5). This adjustment of time spent inhaling may be a method of minimizing the overall amount of time spent at the surface by preventing an oxygen debt or buildup of carbon dioxide. However, the whales may be foregoing full recovery and respiratory gas level readjustments prior to dives with the longest durations and highest lunge counts. Here, they may be incentivized to postpone complete recovery while maximizing foraging time when dense prey patches are available, thus inhaling for durations similar to surface sequences prior to low duration and moderate foraging dives (Boutilier, Reed & Fedak, 2001). This planning prior to foraging has been observed in other cetaceans, such as Risso’s dolphins (Grampus griseus), when prey information obtained from a previous dive was used as a foraging plan for the upcoming dive (Arranz et al., 2018). Pre-planning has been seen in other marine mammals, including the California sea lion (Zalophus californianus), where the extent of lung collapse was directly related to maximum dive depth, indicating larger terminal breaths before deeper dives (McDonald & Ponganis, 2012). Pre-planning has also been observed in other taxa such as penguins. Emperor penguins increase the diving gas volume before deeper dives, and king (Aptenodytes patagonicus) and Adélie penguins (Pygoscelis adeliae) regulate diving gas volume to improve the effects of buoyancy (Sato et al., 2002; Sato et al., 2011). As noted, baleen whales generally feed in prolonged bouts related to diel patterns of prey availability (Friedlaender et al., 2013; Friedlaender et al., 2016), and likely are integrating information about the timing of these events when foraging to postpone full recovery of oxygen stores to periods of rest or non-foraging.

IA and maximum nares area are relative measurements and largely dependent upon the unique tag placement on each whale, thus the values must be normalized to the largest breath to be comparable. In addition to the limited comparison abilities of the IA it was also a time intensive metric to measure, which is why we recommend the use of inhalation duration and the maximum nares expansion over the IA. Additionally, because of the functional relationship (Eq. (1)) between IA and inhalation duration across all whales and breath types, and the significant relationship between maximum nares expansion and inhalation duration, we assume that inhalation duration is a good proxy for nares expansion across breath types (Fig. 4). Inhalation duration, IA, and maximum nares area all depend upon high quality footage, which is an obstacle that should be considered prior to attempting to replicate this method. We recommend further exploration of inhalation duration and maximum nares area to understand the potential as useful metrics that could be used alongside breath count, though this rests on the assumption of near-constant inspiratory flow rates, and individual and species level variation that must be considered (Kooyman, Norris & Gentry, 1975; Sumich, 2001; Fahlman et al., 2016).

Species level variation was observed when examining nares area over the course of an inhalation. Antarctic minke whales rapidly expanded their nares to maximal values, briefly remained at maximal expansion, then rapidly contracted their nares (Fig. 2). Humpbacks rapidly expanded their nares, remained at maximal expansion for a larger proportion of the inhalation duration relative to Antarctic minke whales, then rapidly contracted their nares. This distinction suggests that minke whales spend less time at the surface with fully expanded nares relative to a humpback whale, a much larger mysticete. To compensate for this difference, minke whales may exhibit faster flow rates relative to larger whales and have higher breathing frequencies (Piscitelli et al., 2013); (Blawas et al., 2021). Species level distinctions were also observed when comparing the maximum nares area to inhalation durations (Fig. 4). Higher variation was observed for Antarctic minke whales relative to humpback whales. Given the short amount of time minke whales spend at maximal expansion, relating inhalation duration to nares expansion may be subject to higher rates of error or variation. Much less variation was observed for humpback whales, which suggests a stronger relationship between the magnitude of nares expansion and inhalation durations. Thus, using inhalation duration as a future method to approximate breath size may be more appropriate for larger whales given that the relevant validation experiments are conducted.

We observed that the inhalation duration and nares area across breath types showed similar trends as previous direct measurements of flow rate and changing lung volumes (Kooyman & Cornell, 1981; Fahlman et al., 2015; Fig. 2). With future validation experiments, these metrics, in addition to maximum nares expansion, could improve estimates of tidal volume and of free-swimming large whale energetics, a group for which it is challenging to measure the relationship between inhalation duration and tidal lung volume. Inhalation duration, IA, and maximum nares expansion, however, can be used to improve the accuracy of breath count based conclusions. Improvements in bio-logging technology, such as range-finding could be used to determine the distance to the nares and correct the IA and maximum nares expansion to absolute units, which could enable future research to further utilize the inhalation duration metric and estimate lung volume on a dive-by-dive basis. This could lead to more accurate oxygen consumption estimates for free-ranging baleen whales, which will contribute to important new insights regarding cetacean energetics, foraging performance, response to environmental change, and human disturbance.

Conclusions

Using animal-borne video bio-loggers, this study developed a method that identified considerable breath-to-breath variability by measuring inhalation duration and nares expansion of humpback and Antarctic minke whales. This variation in magnitude was especially apparent in the final breath taken prior to a dive. Our results did not identify a relationship between the total inhalation duration and the previous dive’s duration or foraging effort, though further work should be done to measure the duration and nares expansion of exhalations to fully explore how ventilation metrics change following dives of varying levels of activity. Total inhalation duration did, however, correlate with the upcoming dive’s duration and foraging effort. Our results suggest that as activity and dive duration and lunge count change from low to moderate, the whales adjust their time spent inhaling appropriately, but up to a point. Prior to dives of the longest durations and highest lunge counts, whales may be foregoing complete recovery to take advantage of potentially highly rewarding foraging opportunities. With further validation, inhalation duration and nares expansion could be used to improve estimates of wild cetacean energetics relative to traditional methods alone (e.g., breath frequency). Distinguishing breath types and associating them with certain dive parameters will contribute to assessing energy-conserving mechanisms for cetaceans and other large marine vertebrates. These energetic outcomes can serve as references for improving predictions regarding eco-physiological challenges presented by climate change and anthropogenic disturbance.

Supplemental Information

Supplemental Information 1 Boxplots across integrated area resolutions

Boxplot showing the IA for a resolution of 1, 2, or 3 for two Antarctic minke whales. Resolution of 1 represents IA measurements which were calculated using every frame of the inhalation. Resolution of 2 represents IA measurements which were calculated using every 2 frames of the inhalation. Resolution 3 represents IA measurements using every 3 frames of an inhalation. Boxplot shows mean, standard error, maximum, and minimum.

Click here for additional data file.

Supplemental Information 2 Normalized maximum nares area vs. inhalation duration

Normalized maximum nares area plotted against inhalation duration for all whales in the study. The black line represents the trend line. Observations were grouped by breath type (initial, middle, or terminal) and separated by whale field ID. The labels for each panel represent the species (mn = humpback, bb = Antarctic minke whale) and the field identification number. The y-axis was normalized to the largest IA for each animal.

Click here for additional data file.

Supplemental Information 3 Breath count vs. the previous or upcoming dive duration and lunge count

The breath count surface interval plotted against the upcoming dive duration. The colored lines are separated lunge count category defined separately for humpback and minke whales. For minke whales: non-foraging = 0 lunges; moderate = 1–4 lunges; high = g5 lunges. For humpback whales: non-foraging = 0 lunges; moderate = 1–2 lunges; high = g3 lunges. No model was plotted for high lunges for humpback whales due to small sample size. The lines represent the first degree polynomial GLMM where both lunge count and dive duration of the previous and upcoming dive were predictor variables, breath count was the response, and whale ID and species were random effects. Observations were grouped by species. The labels for each panel represent the species (mn = humpback, bb = Antarctic minke whale).

Click here for additional data file.

Supplemental Information 4 GLMM summary statistics

Statistics for model fit (AIC scores and log likelihood), fixed effects inference (p-value), effect size, and confidence intervals for each GLMM fit in this study. Effect size were calculated using the slope at the model’s median. Confidence intervals were calculated using the Wald method.

Click here for additional data file.

Supplemental Information 5 Video tag data example of exhalation to inhalation transition

Video clip capturing the shape change of the nares as they transition from an exhalation to an inhalation. This breath is a middle breath of the humpback whale mn180607-44. We recommend viewing the transition between the exhalation to the inhalation in frame-by-frame mode to better display how the nares change from an exhalation to an inhalation.

Click here for additional data file.

Supplemental Information 6 Primary dataset for inhalation duration, IA, maximum nares area, and dive metrics

Click here for additional data file.

Supplemental Information 7 Frame area and frame timestamps per inhalation

Click here for additional data file.

Supplemental Information 8 Flow rate over inhalation duration

This data was created using the figure provided in (Sumich, 2001).

Click here for additional data file.

Supplemental Information 9 Complete surface sequence data

This dataset is similar to File S1. This dataset, however, only includes complete surface sequences where every inhalation was recorded and was used for the creation of Fig. 5.

Click here for additional data file.

Supplemental Information 10 Code for study analysis

Click here for additional data file.

We thank the team of people both in Monterey Bay and the Antarctic Peninsula who collected the data that made this study possible. We also thank John Calambokidis from the Cascadia Research Collective, Julian Dale from Duke University, and David Johnston from Duke University for their support in data collection and writing the grants for funding acquisition. We would also like to thank Cade Mirchandani and Philippine Chambault for their assistance in the statistical analysis.

Additional Information and Declarations

Competing Interests

Author Contributions

Animal Ethics

Data Availability

The authors declare there are no competing interests.

Emily C. Nazario conceived and designed the experiments, performed the experiments, analyzed the data, prepared figures and/or tables, authored or reviewed drafts of the article, and approved the final draft.

David E. Cade conceived and designed the experiments, performed the experiments, analyzed the data, prepared figures and/or tables, authored or reviewed drafts of the article, and approved the final draft.

K.C. Bierlich performed the experiments, authored or reviewed drafts of the article, and approved the final draft.

Max F. Czapanskiy performed the experiments, analyzed the data, prepared figures and/or tables, authored or reviewed drafts of the article, and approved the final draft.

Jeremy A. Goldbogen performed the experiments, authored or reviewed drafts of the article, and approved the final draft.

Shirel R. Kahane-Rapport conceived and designed the experiments, performed the experiments, authored or reviewed drafts of the article, and approved the final draft.

Julie M. van der Hoop conceived and designed the experiments, authored or reviewed drafts of the article, and approved the final draft.

Merceline T. San Luis performed the experiments, authored or reviewed drafts of the article, and approved the final draft.

Ari S. Friedlaender conceived and designed the experiments, performed the experiments, analyzed the data, prepared figures and/or tables, authored or reviewed drafts of the article, and approved the final draft.

The following information was supplied relating to ethical approvals (i.e., approving body and any reference numbers):

The humpback tag deployments were based in the Monterey Bay National Marine Sanctuary under the NMFS permit #20430. The Antarctic minke whale tags were deployed in the West Antarctic peninsula under NMFS permit #23095. This study was also conducted under the UCSC IACUC permit Friea1706 and the ACA permit 2015-011.

The following information was supplied regarding data availability:

The raw integrated area measurements and associated data, additional sources of data that were used to provide the figures and the raw code used to complete the analysis are available in Supplementary Files.

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
