# Peer review of "Baleen whale inhalation variability revealed using animal-borne video tags"

_PeerJ, doi:10.7717/peerj.13724_

## Round 0.1 · original submission · Minor Revisions

Both reviewers appreciate this paper, and have offered a few minor suggestions to improve clarity.

·

Basic reporting

Review for ms 73369

I feel this is a useful and timely study. With the exceptions listed below, the basic reporting is good, as are the experimental design and presentation of results. I must state that judging the validity of the statistical methods used should be done by one with a stronger statistical background than mine.

Experimental design

no comment

Validity of the findings

no comment

Additional comments

Specific comments:

Lines 97ff, 165ff, 181: With the large number of variables discussed in this paper, some abbreviated, some not, this paper would benefit from the addition of a concise table of variables and their consistently used abbreviations (e.g. Fahlman et al, 2015; Sumich, 2021).

Line 142ff: Establishment of the beginning of each inspiration is critical to this study. Although I find no problem with the approach stated here, are there associated audio tracks that might provide confirming acoustic ventilatory flow signals for the transition from expiration to inspiration.

Line 150: This seems an appropriate spot to include the terms inter-breath-interval (IBI) and extended dive times (EDT) and add a table of summary statistics of IBI and EDT values for each whale (See Butler and Jones, 1997 for background).
The selection of <5s duration and <10m depth to define surface sequences seems arbitrary. Are there good reasons for choosing these values, and were alternative approaches considered, for example examining the bimodal distributions of log transformed values of all IBI and EDT dive durations of each whale and using the intermediate minimum (likely around 20-30s) to define the boundary between IBIs and EDTs.

Line 169: Does the video frame rate (30Hz) define the resolution of the time scale for determining expiration duration?

Line 181: This is the first occurrence of IA as the abbreviation for the variable “integrated area’ although the variable name is used without abbreviating in the previous paragraph. See first comment above.

Line 224: …duration.,

Lines 256, 322, 327-328: Sumich, 2001 and Sumich and May, 2009 only measured expiration durations on gray whales, not inspirations of odontocetes.

Lines 273-287: Why is Fig. 5 on line 273 placed prior to Fig. 4 on line 287.

Line 294: …confirmed that inhalation…

Line 364: …which could indicate suggests that…

Lines 414ff: excellent explanation

Line 425: Sumich et al, 2001
Line 444-445: delete entire first sentence of Conclusion, as that statement does not apply to all previous studies.
Line 455ff: Favilla and Costa, 2020 have shown that ADL of blue and fin whales is about 5x their typical foraging EDT values. I suggest some discussion is needed here to explain why inspiration duration (as a proxy for O2 loading) should correlate with O2 requirements of anticipated upcoming dives when a whale’s upcoming typical dive durations only uses 1/5th of its actual aerobic dive capacity.
Fig. 1: inset graph needs units, otherwise very good.

Fig. 4. This fig. is visually confusing to me. It seems to me to indicate that total inhale duration begins to decrease for upcoming dive durations longer than about 300s. I don’t see that from the distribution of data points. Why might that be and do you think it is physiologically meaningful?

Reviewer 2 ·

Basic reporting

Overall this is a well written paper and a creative approach to addressing the methodical challenges of measuring energetic requirements of large whales. The introduction is concise and clear, citing relevant literature and the authors clearly state the goals of the project.
There are a few sections in the methods and discussion that are unclear and the authors should make some revisions to help clarify their statistical analysis and discussion points. Details are provided below in other sections.

Experimental design

The authors clearly state the research questions and why the study is needed. This is a great use of data collected for other studies. They clearly defined what criteria the video must meet to be included and I appreciate how they performed tests on a subset of the data to defend their methods (only measuring one side, etc). One point I would like to see – is how hard is it to get this data – did they tag 100 whales with video tags and there were only 6 useable datasets? How many hours did they use for each whale (could be added to Table 1). That would allow the reader to assess if this is a method that can be relied on. They also talk about how need more validation. How will you validate this for large whales?
The statistical analysis section is a little confusing and the authors should provide more information on model validation.
Line 218-236: I am a little confused by this section. I thought you normalized the data so you could compare the whales, but here you say you normalize and then test separately? Then one sentence later you say given the individual variation you “also” plotted separately. You need to make it clearer why you normalized if you still analyzed separately. But when I look at your stats and results, you use GLMM with whale ID as random effect – so then you did not analyze separately like you stated above? I suggest revising this so it is a little easier to figure out what you did. Also – there seem to be some difference between species – I am not sure it is appropriate to keep them both in the same model? At a minimum you need to justify why you think this is okay.
Line 238 – 245: I like that you linked behavior to inhalation duration, but I thought a part of this would be linking things like IA and max nares expansion. If you say those vary, especially for terminal breath, I thought you would want to link this to activity (at the end of the discussion I see you justified why you just did duration – but you might want to suggest that here because I felt like it was missing until the very end). Also if you are only including duration, would it be that different than if you just counted respirations? Could you also look at relationship between # of respirations to dive parameters and then maybe compare? – does duration give you better results than # of respirations? The variation in inhalation duration is much smaller in the minkes between all breath types so maybe for this species respiration count would be just as useful as total surface inhalation duration. The comparison might make a stronger argument for why this much more time consuming technique is a better method – at least in some species.

Validity of the findings

In general, I think the authors conclusions were appropriate for their findings. However, I would like them to provide more information about their models. This is important to include so the readers can assess their conclusions. While they provide p-values, they do not provide any information on effect size, confidence intervals or model fit which are all necessary to understand what the significant finding mean.
On line 368-375, when the authors discuss how measures of IA and maximum expansion may improve estimates of body density, drag, etc., I felt like they were overreaching a bit. They don’t make it clear how variation they found will help with these measurements. There are still many issues to overcome such as tag placement. In general, this superficial discussion point came off a bit like a tangent and overreach. I think the discussion would be stronger without it.
Paragraph 386- 412: The authors comment on longer surface inhalation durations during intermediate dives. The explanation of this pattern is a bit weak. They talk about how may manage on level of a bout, but it is not clear why it would be shorter after long dives (they do a better job of clarifying in the conclusions). It would also be helpful to see more model results. I did not find Fig 4 very convincing –there is a lot of variation in the data both within and between individuals so would be good to have information on effect size and confidence intervals.
Line 424-425: The authors mention species may differ – so is it appropriate to put both in the same models? That is what your methods and Table 3 results suggests you did.

Additional comments

I enjoyed reading this manuscript and appreciate the creative approach. Below are a few small suggestions to improve the readability.

Abstract
Line 28: Don’t need word historically.
Line 33: don’t need word additional (this would be the first free-ranging correct?)
Line 38: Don’t you think you need the word physiological.
Introduction
Line 62: Pederson reference I think is spelled wrong and not in ref list.
Line 69: respiratory physiological does not make sense – do you mean respiratory physiology?

Methods:
109 – 113: Awkward and a little repetitive first sentence. Try to rewrite so more direct or break in two sentences - Maybe more like: This study used data from video-recording motion sensory tags from Customized Animal Tracking Solutions (CATS) that were deployed for other projects (citations).
Line 113-115 – also a bit awkward. You have a very complex subject – suggest revising so verb is earlier in the sentence. Maybe something like: Whales were selected if the video tag was placed directly behind and facing the nares, the nares were clearly visible, and there was at least 2 hrs of video where the tag did not shift position.
Line 120: Don’t think you need word “based”
Results:
Line 248 – 249: I find this sentence a little vague – what do you mean by timing did not vary? – duration? Because the next sentence highlights a difference in timing – when they reached max expansion. Be a little clearer about what you mean or state that they did vary slightly?
Discussion
Lines 319-325: I don’t think the comparison to flow rate is as clear as the authors think – or I am missing it. Flow rates are high very early, and nares are just starting to expand when flow rate is already high. Maybe explain a bit better so easier for reader to grasp.

---

## Round 0.2 · accepted · Accept

Thanks for your work on the revision; the reviewers and I think that you have done a great job incorporating their suggestions.